# The Effects of Protease Supplementation and Faba Bean Extrusion on Growth, Gastrointestinal Tract Physiology and Selected Blood Indices of Weaned Pigs

**DOI:** 10.3390/ani12050563

**Published:** 2022-02-23

**Authors:** Anita Zaworska-Zakrzewska, Małgorzata Kasprowicz-Potocka, Klaudia Ciołek, Ewa Pruszyńska-Oszmałek, Kinga Stuper-Szablewska, Andrzej Rutkowski

**Affiliations:** 1Department of Animal Nutrition, Faculty of Veterinary Medicine and Animal Science, Poznań University of Life Sciences, ul. Wołyńska 33, 60-637 Poznań, Poland; anita.zaworska-zakrzewska@up.poznan.pl; 2Animal Nutrition Research Station in Gorzyń, Poznan University of Life Sciences, ul. Wojska Polskiego 28, 60-637 Poznań, Poland; klaudia15@gmail.com; 3Department of Animal Physiology and Biochemistry, Faculty of Veterinary Medicine and Animal Science, Poznań University of Life Sciences, ul. Wołyńska 35, 60-637 Poznań, Poland; ewa.pruszynska@up.poznan.pl; 4Department of Chemistry, Faculty of Wood Technology, Poznań University of Life Sciences, ul. Wojska Polskiego 75, 60-625 Poznań, Poland; kinga.stuper@up.poznan.pl

**Keywords:** digestibility, extrusion, faba bean, protein, protease, pigs

## Abstract

**Simple Summary:**

Faba beans could be a valuable raw material in pigs’ diets, but the presence of anti-nutritional factors limits their wide use. The aim of the study was to investigate how the extrusion of faba bean seeds and/or the addition of protease to pigs’ diets affected the animals’ growth parameters, digestibility of nutrients, selected physiological parameters of the digestive tract, and biochemical blood parameters. Our research showed that extrusion increased the nutritional value of faba bean seeds, especially by reducing antinutritional factors, but in comparison with raw seeds, it did not improve the pigs’ growth performance, digestibility of nutrients, intestinal structure, and physiology when the content of faba beans in the diet was below 10%. Thanks to protease supplementation in our study, protein and oil levels in the diet were reduced while maintaining the same pigs’ performance, which cut the cost of feeding. The extrusion and enzyme additives did not improve the pigs’ growth performance in this experiment, but protease appears to be highly promising in the commercial nutrition of pigs.

**Abstract:**

The aim of the study was to investigate how the extrusion of faba bean seeds (var. Albus) and/or the addition of protease to pigs’ diets affected the animals’ growth parameters, digestibility of nutrients, selected physiological parameters of the digestive tract, and biochemical blood parameters. A 28-day experiment was conducted on 32 pigs weighing 9 ± 0.2 kg. The animals were allocated to four treatments in a 2 × 2 factorial arrangement with the main effects of extrusion (raw or extruded) and effects of protease supplementation (0 and 0.05%). Extrusion reduced the levels of neutral detergent fibre, trypsin inhibitor, phytate-P, and resistant starch but did not improve the digestibility of protein and dry matter in faba bean seeds. The pigs’ growth performance, ileal digestibility, enzyme activity, and morphometric parameters of the ileum were not significantly affected by extrusion, except for a higher feed intake between the 15th and 28th day of the experiment. The protease supplementation gave comparable results as the diet without protease, except the feed conversion ratio (in the periods of 15–28th day and 0–28th day), which was higher than in the groups without protease. The extrusion and protease increased acetate and acetoacetate contents in the cecal digesta, but propionate, butyrate, and isovalerate concentrations in the digesta of the pigs in this group were lower. Thanks to protease supplementation, protein and oil levels in the diet were reduced, which cut the cost of feed mixtures. The extrusion and protease additive combined together did not improve the pigs’ growth performance in this experiment.

## 1. Introduction

In modern farming systems, balanced diets meeting nutritional requirements at each growth stage of pigs are essential for them to achieve full genetic potential. Researchers are particularly interested in the use of native crops and products as alternative sources of protein and fat in pig farming. In recent years, there have been several studies on feeding young pigs with legumes [1,2,3]. Most of them showed that legume seeds still have a lower nutritional value than soybean meal (SBM) because they contain antinutritional factors (ANFs) [4,5,6]. Faba bean seeds (*Vicia faba*) are a good source of protein, but have insufficient amount of sulphur-containing amino acids (AAs); relatively high content of fibre; and some ANFs such as tannins, phytate, and protease inhibitors [1,7,8,9]. Although currently low-tannin cultivars are used, the nutritional effect of faba bean seeds is still lower than that of SBM; thus, various treatments are used to decrease the content of ANFs [1,2,8,9,10].

Extrusion is continuous cooking under pressure, moisture, and elevated temperature [11]. This process may be applied to improve the absorption and utilisation of feed nutrients by increasing the digestibility of protein, amino acids, energy, and starch and to increase the sterility of feed [1,8,12,13,14]. Extrusion can also reduce the levels of some ANFs found in legumes such as tannins, phytic acid, and trypsin inhibitors (TIA) [1,3,8,9,15,16,17]. Nonetheless, studies published so far have shown that the effect of extrusion is variable. O’Doherty and Keady [16] found that the extrusion of peas resulted in a noticeable decrease in the tannin content, whereas Hejdysz et al. [8] observed that the content of TIA in pea seeds decreased but the tannin content increased slightly after extrusion. In the last ten years there have been numerous articles on the extrusion of bean, pea, and lupin seeds but rather few studies on faba beans [18]. Some authors found that the negative effects of ANFs are evinced by damage to intestinal structures and absorptive surface area and changed the time of transit through the gastrointestinal tract and insufficient secretion of endogenous proteases [19]. As a result, nitrogen and other undigested nutrients could be freed into the environment. The use of enzymes such as proteases could effectively improve the utilization of protein and with ever-increasing feed prices, reducing the cost. It could also help to extract more nutrients from feed ingredients and consequently lead to an improved growth performance in animals. The concept of using exogenous microbial enzymes in the animal feed industry has been established to improve the nutritive value of feed ingredients. This can be achieved through the displacement of expensive protein feed material in the diet by assigning the nutrient matrix on the enzyme in a less expensive formulation [20]. Zuo et al. [14] found that protease supplementation increased the growth performance of weaned piglets and improved their intestinal development, protein digestibility, nutrient transport efficiency, and health. Similarly, Hanczakowska and Świątkiewicz et al. [21] found that the protease additive could improve the performance of fattening pigs fed with faba bean seeds. Therefore, the authors of this study hypothesised that extruded faba beans with protease could be used to partly replace SBM in mixtures for pigs and successfully improve the growth of weaning pigs.

The aim of this study was to determine if the extrusion cooking of faba beans and the supplementation of the nutrient matrix with protease may affect the growth parameters of pigs, the digestibility of nutrients, and the selected physiological parameters of the digestive tract and biochemical blood parameters.

## 2. Material and Methods

### 2.1. Faba Bean Seeds

White-flowered, narrow-leaved, low-tannin faba bean seeds (*Vicia faba L.* var. Albus) acquired from the Plant Breeding Station in Strzelce, Poland, were used in the experiment. The samples of faba bean seeds came from crops harvested in 2017. Faba bean seeds were extruded with a KMZ 2 extruder (Moscow, Russia) (500 kg/h) under the following conditions: moisture—about 21%; exposure time—10 s; temperature—135 ± 10 °C; pressure—30 kg × cm^2^. The extruded faba bean seeds were cooled down to room temperature, ground with a laboratory grinder with a 1.00 mm sieve (Retsch, Haan, Germany), and stored at 4 °C. The chemical compositions of raw and extruded faba bean seeds can be found in the Results section (Table 1).

### 2.2. Animals, Diets, and Protease

All experimental procedures applied in our experiment followed the guidelines of Directive 2010/63/EU of the European Parliament and of the Council on the protection of animals used for scientific purposes [22]. The experiment was approved by the Local Ethical Committee in Poznań—Resolution No. 43/2011 of 15 May 2011.

The experiment was conducted on 32 castrated male pigs (Naïma × (Pietrain × Duroc)) aged 42 days with an initial body weight (BW) of 9.0 ± 0.2 kg. Before the experiment, the animals were housed for five days in straw, which had been removed before the beginning of the experiment. Young pigs were randomly allocated to one of four dietary treatments and kept in individual cages for 28 days. Raw or extruded faba bean seeds with or without protease supplementation used in the diets were the factors of the experiment conducted in a 2 × 2 design. The level of crude protein (CP) in the enzyme treatments was about 5% lower, according to the protease matrix recommendation. All diets were offered as a mash ad libitum. In order to calculate digestibility coefficients, 0.3% of titanium dioxide was added to all the diets. The complete diets for the experiment were formulated according to the recommendations of the Recommendations for the Energy and Nutrient Supply for Pigs (GfE) [23], as shown in Table 2.

Protease supplementation was applied at a dose of 0.5 g/kg of the experimental diet. It contained heat-stable protease (min. 600,000 U/g, EC 3.4.21.19) from viable spores of *B. licheniformis* (min. 1 × 10^9^ CFU/g, ATCC 53757). The diets were supplemented with protease to reduce the amount of crude protein and percentage of some amino acids as well as the content of metabolisable energy by decreasing the amount of SBM in the diets (diets II and IV—Table 2). The mean daily weight gains (DWG) and feed intake (FI) were recorded to calculate the mean feed conversion ratio (FCR).

After the last day of the experiment, the pigs received their last meal one hour before they were stunned by electric shock. The period between feeding and euthanasia lasted 3 h ± 15 min. Immediately after euthanasia, blood samples were collected from all animals in each group (*n* = 8), about ten minutes after the euthanasia ileum samples were collected for morphometric analyses. Duodenal content was collected into plastic bags and stored at −80 °C for pancreatic enzyme analysis. The samples were collected from the ileum (about 50 cm before the ileocecal valve). pH values and ammonia content were measured in the samples of ileal and cecal digesta and calculated per dry matter (DM). The viscosity of the ileal digesta was also analysed. The apparent ileal digestibility (AID) coefficients were calculated from the following equation [24].
(1)$$\mathrm{AID}=100-\left[\left(\frac{{\mathrm{TiO}}_{2}\frac{g}{\mathrm{kg}}\mathrm{diet}\;}{{\mathrm{TiO}}_{2}\frac{g}{\mathrm{kg}}\mathrm{digesta}/\mathrm{excreta}}\right)\times \left(\frac{\mathrm{nutrient}\frac{g}{\mathrm{kg}}\mathrm{digesta}/\mathrm{excreta}}{\mathrm{nutrient}\frac{g}{\mathrm{kg}}\mathrm{diet}}\right)\right]$$

### 2.3. Blood Sampling

Blood samples were collected into Vacutainer Serum Separator Tubes (BD SST II Advance, Franklin Lakes, NJ, USA). The samples were incubated for 15 min at room temperature to clot and then they were centrifuged for 10 min at 3500× *g* at 4 °C. Serum was transferred into new tubes, immediately frozen, and stored at −80 °C until future analysis.

### 2.4. Metabolic Blood Serum Profile

The metabolic profile was determined with commercially available colorimetric assays according to the manufacturer’s instructions. The optical density of the samples was determined with a microplate reader (Synergy 2, Biotek, Shoreline, WA, USA). The following blood serum parameters were measured by means of Pointe Scientific assays (Pointe Scientific, Canton, MI, USA): glucose (Cat. No.: G7519), triglycerides (TG; Cat. No.: T7531), total cholesterol (TCh; Cat. No.: C7510), high-density lipoprotein (HDL; Cat. No.: 7545), low-density lipoprotein (LDL; Cat No.: 7574-D) fractions, albumin (Cat. No.: A7502-500), urea nitrogen (BUN/UREA; Cat. No.: B7552-120), creatinine (Cat. No.: C7539-500), the activity of alanine aminotransferase (ALT; Cat. No.: A7526), aspartate aminotransferase (AST; Cat. No.: A7561), γ-glutamyltransferase (GGT; Cat. No.: G7571), and alkaline phosphatase (ALP; Cat. No.: A7516-120). The level of non-esterified fatty acids (NEFA) was measured with a WAKO kit (Cat No.: 434-91795 and 434-91995; Wako Chemicals, Richmond, VA, USA). The total protein concentration was measured with an Alpha Diagnostics kit (Cat. No.: A6502-100; B6528-125).

### 2.5. Enzyme Activity

Immediately after collection the duodenal contents were frozen in liquid nitrogen and stored at −80 °C until analysis. The activities of digestive enzymes (amylase, lipase, trypsin, chymotrypsin, and maltase) were determined with the method described by Pruszyńska-Oszmałek et al. [25]. Frozen duodenum contents were weighed and briefly homogenised in an appropriate volume of Tris Buffered Saline (TBS, pH 7.4) by means of a TissueLyser II homogeniser (Qiagen, Germantown, MD, USA) and vortex. Next, the homogenates were centrifuged at 10,000× *g* for 16 min and the supernatants were transferred into new tubes for future analysis (all procedures were performed in ice). The activities of trypsin and chymotrypsin were analysed with commercially available colorimetric assay kits from BioVision (Milpitas, CA, USA). The activities of amylase and lipase were analysed with Pointe Scientific assay kits (Pointe Scientific, Canton, MI, USA). Maltase activity was determined by measuring the increase in absorbance at 400 nm caused by the hydrolysis of p-nitrophenyl-α-D-glucopyranoside (PNPG) (Dahlgvist, 1984). The absorbance of the samples was measured with a Synergy 2 microplate reader (Biotek, Shoreline, WA, USA). Protein content was measured with a BCA assay protein kit (Thermo Scientific, Waltham, MA, USA).

### 2.6. Chemical Analysis

For chemical analyses, representative samples of raw and extruded faba bean seeds were ground to pass through a 0.5 mm sieve. Faba bean seeds (diets and ileum digesta) were analysed in duplicate for DM, CP, ether extract (EE), crude fibre (CF), crude ash (CA), acid detergent fibre (ADF), neutral detergent fibre (NDF), calcium, and phosphorus with methods at 934.01, 976.05, 978.10, 942.05, 973.18, 984.27, and 965.17, respectively, according to AOAC [26]. The starch content in faba beans was measured with a diagnostic assay kit for the agricultural industries (Megazyme International; AOAC [26]: method 996.11) using thermostable α-amylase and amyloglucosidase. Resistant starch (RS) content was analysed with a resistant starch assay kit (Megazyme International, Wicklow, Ireland; AOAC [26]: Method 2002.02) with modified incubation time [27,28]. Tannin content in the faba bean samples was analysed with the method described by Kuhla and Ebmeier [29]. The TIA content was measured according to PN EN ISO 14902:2005 [30]. Raffinose family oligosaccharides (RFOs) were extracted and analysed by means of high-resolution gas chromatography as described by Zalewski et al. [31]. Phytate content was measured according to the method described by Haug and Lantzsch [32]. The non-starch polysaccharide content was measured by means of gas-liquid chromatography (neutral sugars) and colorimetry (uronic acids). The procedure invented by Englyst and Cummings [33,34] with some modifications [35] was used for neutral sugars. The procedure described by Scott [36] (1979) was used to measure the content of uronic acids. The content of amino acids was measured with an AAA-400 Automatic Amino Acid Analyser (Ingos Ltd., Prague, Czech Republic), using ninhydrin for postcolumn derivatisation. Before analysis, the samples were hydrolysed with 6 M HCl for 24 h at 110 °C (procedure 994.12; AOAC [26]. Titanium dioxide content was estimated with the method described by Short et al. [37], and samples were prepared according to the procedure described by Myers et al. [38]. pH was measured with a microelectrode and a pH meter (model 301, Hanna Instruments, Vila do Conde, Portugal). Ammonia was extracted and analysed with the spectrometric method using Nessler Reagent (POCh, Gliwice, Poland). In order to measure the digesta’s viscosity, the samples were centrifuged at 10,000× *g* for 10 min at 4 °C. The supernatant was withdrawn, and viscosity was measured with a Brookfield Digital DV-II + cone/plate viscometer (Brookfield Engineering Laboratories, Stoughton, MA, USA) maintained at 40 °C at a shear rate of 60 s. Water extract viscosity units are mPas (mPa·s = cP = 0.01 dyn/s/cm^2^). The content of short-chain fatty acids was measured with the method described by Stuper-Szablewska et al. [39].

### 2.7. Morphometric Analysis

Before morphometric analysis of the ileum tissues, parts of the ileum were fixed in 4% formalin buffered with CaCO_3_ solution, then washed and dehydrated in ethanol of increasing concentration, X-ray xylene, and then embedded in paraffin. Sections with a thickness of 10 µm were cut on a rotary microtome (Thermo Shandon, Runcorn, UK) and then placed on microscope slides coated with chicken egg whites with the addition of glycerin. Specimens were deparaffinised, rehydrated, and then stained with the PAS (Periodic acid-Schiff) technique by using the Schiff reagent for intestinal morphometric analysis. A Nikon Ci-L microscope integrated with a Nikon DS-Fi3 camera and NIS Elements software (Nikon Instruments Inc., Melville, NY, USA) was used to measure the thickness of the muscular membrane, height and width of villi, and depth of intestinal crypts (in ten replicates).

### 2.8. Statistical Analysis

Data were analysed as a 2 × 2 factorial experiment using the general linear model procedures of SAS 9.2 (SAS Institute, Cary, NC, USA), with a model including the main effects of extrusion and the effect of protease, as well as their interaction. Means were compared with Duncan’s range test to determine significant differences between the means with a significance level of *p* < 0.05. Student’s *t*-test was applied to compare the chemical composition of raw and extruded faba bean seeds.

## 3. Results

### 3.1. The Chemical Composition of Faba Bean Seeds

The chemical composition of raw and extruded faba bean seeds is shown in Table 1. The mean CP concentration in the four samples of raw faba beans was 34.7% DM and did not differ significantly from the mean CP concentration in the extruded seeds. The extrusion cooking of faba bean seeds also had no influence on the content of ADF, total starch, and all amino acids and ANFs, except for TIA and phytic acids. The extruded faba bean seeds also had a significantly lower (*p* ≤ 0.05) NDF concentration and RS level (≤0.001), which were, respectively, about 35 and 95% lower than in the raw seeds. The analysis also showed that the extrusion had no influence on the content of oligosaccharides, tannins, or non-starch polisaccharides (NSP). However, it significantly decreased TIA and phytic-P contents to about 67% and 62%, respectively, as compared with the raw faba bean seeds.

### 3.2. Animal Experiment

During the entire experiment, the pigs were in good health, and there were no mortalities or visible disease symptoms, including diarrhoea. Their growth performance results are shown in Table 3. Extrusion cooking and protease supplementation had no effect on the pigs’ performance, except for DFI in the second period (15–28 days) as well as FCR in the second period and during the entire experiment. The DWG of the pigs fed with extruded faba bean seeds tended to increase. Protease supplementation via displacement of soybean meal from the diet provided comparable results, except FCR in the periods of 15–28 days and 0–28 days, (*p* ≤ 0.01 and *p* ≤ 0.05, respectively), which was higher than in groups without protease. There were no interactions between experimental factors.

The dietary inclusion of extruded faba bean seeds did not significantly (*p* > 0.05) affect the apparent ileal digestibility coefficients of CP and DM in the diet (Table 4). Protease supplementation provided comparable results as in groups without the enzyme. No interactions were found.

The effect of diets on physicochemical indicators of the gastrointestinal tract is shown in Table 5. The liver weight, liver weight to BWG ratio, ammonia content, and viscosity in the digesta were not affected by extrusion or protease supplementation. The extrusion increased the pH value of the cecal digesta (*p* ≤ 0.01) and decreased the dry matter content in the ileal digesta (*p* ≤ 0.004). Likewise, there was no interaction between experimental factors.

Extrusion significantly increased (*p* ≤ 0.05) the content of acetic and acetoacetic acids in the cecal digesta but reduced (*p* ≤ 0.05) the content of propionic, butyric, and isovaleric acids (Table 6). The protease supplementation of the diet significantly (*p* ≤ 0.05) reduced the content of propionic, butyric, and isovaleric acids but increased the acetoacetic acid content in the cecal digesta. There was an interaction between the factors in the valeric acid content.

Neither extrusion nor protease supplementation affected (*p* > 0.05) the activity of enzymes in the distal colon (Figure 1) or the morphological parameters of the ileum (Table 7). There was no interaction between experimental factors.

All blood biochemical indicators met the physiological standards (Table 8). The groups differed significantly in glucose, ALT, and GGT levels. Extrusion significantly increased the blood serum glucose and ALT levels (*p* ≤ 0.05). The highest GGT level was noted in the group which received the diet with raw faba bean seeds; the lowest GGT level was observed in the group which received the diet with extruded faba bean seeds (*p* ≤ 0.043). There was an interaction between the factors in GGT only.

## 4. Discussion

The analysis of the chemical composition showed that the seeds of faba bean cv. Albus contained 34.7% of crude protein and 41.5% of starch in DM basis. They also had a low content of tannins and TIA, which was similar to the values measured for this cultivar by Hejdysz et al. [8]. The seeds were characterised by a high level of resistant starch (RS), which exceeded 18%. Punia et al. [40] observed that the RS content in faba bean seeds ranged from 3.3% to 6.5%. RS is the sum of starch and its derivatives that are not digested in the small intestine of healthy individuals. After the ingestion of legumes, the release of glucose into the bloodstream becomes slower due to RS, which reduces glycaemic responses. However, there could be differences between the results of in vitro and in vivo studies because of the species and age of animals used in the test, the enzyme cocktail, and the time of digestion.

### 4.1. Extrusion Effect

Optimal temperature, pressure, and conditioning during extrusion cooking inactivate ANFs and may enhance the utilisation of nutrients due to physicochemical changes in the structure of seeds [41]. In our study, the extrusion process had an inconsiderable effect on the content of oligosaccharides, NSP, and amino acids in faba bean seeds, but it considerably reduced the content of TIA, RS, and phytate-P by about 67%, 95%, and 62%, respectively. Some authors claim that RS is frequently analysed as the sum of starch and starch degradation products not absorbed in the small intestine. Part of the starch from legume seeds cannot be digested and finds its way to RS. The extrusion process changed the levels of phytate in the faba bean samples. During extrusion, some inositol hexa-phosphate molecules may have been hydrolysed to penta-, tetra-, and triphosphates, reducing phytic P content in extruded seeds. The level of tannins was low and similar in raw and extruded faba bean seeds, but in the final diet, it did not exceed 0.0006%. These results are similar to the data presented by Hejdysz et al. [8], Diaz et al. [42], and Meng et al. [43]. It seems that modern faba bean cultivars with low tannin content can be better accepted as feed components for pigs. Extrusion affected the NDF level, which means that insoluble fibre was converted into soluble fibre during the process. The NDF levels were very similar to the levels of the same cultivar presented in the study by Hejdysz et al. [8].

Similarly to the study by Tuśnio et al. [1], the reduction in ANFs in faba bean seeds by extrusion did not improve the apparent ileal digestibility coefficients of CP and DM in the diets, nor did it improve the activity of digestive enzymes or morphometric parameters of the ileum. The influence of the extrusion of legume seeds on AID was investigated by Zuo et al. [14] and Zaworska et al. [2], who found that it could increase the digestibility of protein and some amino acids. The conditions of extrusion significantly influence the structure of seeds. Hejdysz et al. [8] used seeds extruded under the same process conditions and observed improved digestibility coefficients of some nutrients in broiler chickens. Legumes provide a good example of improved protein digestibility and bioavailability of sulphur-containing amino acids through the thermal unfolding of major globulins and thermal inactivation of TIA. However, extensive loss of lysine may take place when legumes are extruded under severe conditions of temperature or shear forces (>l00 rpm) at low moisture (≤15%), especially in the presence of reducing sugars—the Maillard condensation [43]. It is also important to note that the content of faba beans in experimental diets was only 9.1%; thus, it may have been too low to observe differences in the digestibility coefficients. This fact was also corroborated by the unchanged morphometric parameters of the ileum and the activity of digestive enzymes. Similarly to our study, Tuśnio et al. [1,9] found that, in comparison with the diet with unprocessed legume seeds or soybean meal, extrusion had no effect on the crypt depth and villus height as well as enzyme activity. However, in our study, extrusion affected some other physical parameters in the ileum and cecum, increased the pH value, and caused fluctuations in the concentration of some acids. There was a higher content of acetic and acetoacetic acids in the digesta of the pigs which received extruded seeds. These observations were in line with the results of the study conducted by Biagia et al. [44], who found a negative correlation between the dietary tannin content and the production of acetate, propionate, and butyrate by microflora, but in our study, low-tannin faba beans were used. Tuśnio et al. [9] did not find any changes in the concentration of fatty acids and ammonia in the digesta collected from the cecum and colon when increasing doses of extruded seeds were included in pigs’ diets. It is also interesting that extrusion significantly reduced the dry matter content in the ileum. Extrusion cooking modifies the size of particles, solubility, and chemical structure of various fibre components. This may change bacterial degradation in the intestine and alter the physiological properties of seeds [41].

On the other hand, changes in the physicochemical structure of extruded seeds are believed to improve palatability and digestibility [45], but this may not improve growth performance [46]. In our study, extrusion cooking did not improve the pigs’ growth performance. Only the feed intake in the second period was higher, but the FCR in the second period and during the entire experiment did not improve. Additionally, the excessive consumption of carbohydrates in faba beans, especially α-galactosides, may also cause undesirable effects. These carbohydrates are not hydrolysed by digestive enzymes of monogastric animals and may cause flatulence and diarrhoea in animals, but this problem was not observed in our study. Similarly to this study, the experiment conducted by Tuśnio et al. [1,9] on weanling pigs showed that the extrusion of pea and faba bean seeds did not affect the animals’ feed intake or body weight gains. Ivarsson and Neil [10] found that the seeds of different faba bean varieties resulted in similar DWG, and FI. Zaworska et al. [2] fed pigs with extruded pea seeds and did not observe improvement in their growth performance. Lancheros et al. [47] mentioned some experiments with slight improvements, which may have been caused by the type of extruder used and/or different forms of the diet provided to pigs. The extrusion only affected the concentration of glucose and ALT in the blood serum. The elevated glucose content may have resulted from the reduction of RS in extruded faba bean seeds and higher utilization of sugars by pigs [40].

### 4.2. Protease Effect

After weaning, the activity of digestive enzymes in the pig’s stomach and pancreatic tissue decreases dramatically; thus, it is necessary to include dietary protease to complement endogenous proteolytic enzymes so that they can digest nutrients more efficiently, especially when there is a lower content of digestible protein in the diet. In this study, the protease additive did not improve the digestibility of the dietary nutrients and did not affect the activity of digestive enzymes and the morphometric parameters of the ileum. Świątkiewicz et al. [48] observed that protease supplementation increased the AID of CP in growing and finishing pigs, but DM digestibility increased only in growing pigs and it was lower than in the finishing animals. Zuo et al. [14] found that protease supplementation at a dose of 200 and 300 mg/kg, but not at 100 mg per kg of the diet, improved the digestibility of crude protein but did not improve the digestibility of dry matter. Ma et al. [49] found increased apparent digestibility coefficients of crude protein in the pigs which received dietary supplementation with ‘alkaline’ or ‘acidic’ protease but not ‘neutral’ ones. Zuo et al. [14] observed that the supplementation of a diet containing unprocessed seeds with an enzyme additive improved the morphometric parameters of the ileum and the activity of pepsin, amylase, and trypsin. In our study the protease additive had no effect on the pH, ammonia, or dry matter content in the digesta, but it reduced the concentration of some fatty acids and increased the content of acetoacetic acid only. The effectiveness of protease supplementation of the diet provided to weaned pigs and its mechanisms are still not clear, especially when low-digestible protein sources are used.

The protease supplementation of the pigs’ diet resulted in a comparable performance as in the group fed with the diet without the enzyme, but FCR was higher. Similarly, Hanczakowska and Świątkiewicz et al. [21] and Ma et al. [49] found that although protease supplementation could improve the average daily weight gain, it did not have positive influence on the FI and FCR when the diet contained processed or unprocessed legume seeds. On the contrary, Zuo et al. [14] found that higher dietary protease supplementation increased the growth performance of pigs. It may improve the development of intestines and protein digestibility and increase the activity of enzymes such as pepsin, amylase, and trypsin when weaned pigs are fed with low-digestible protein sources such as raw soybean seeds. Higher nutrient digestibility is caused by the slow development of the digestive function during the growth of pigs. Therefore, the effectiveness of protease may be related to its type and level as well as pigs’ growth stages and diet [14]. Thanks to protease supplementation in our study, it was possible to reduce the level of protein and oil in the feed (expensive soybean meal) while maintaining the same production performance of the pigs, which reduced the cost of feeding.

The analysis of biochemical blood indicators showed that the nutrient matrix and protease supplementation did not change them. Zuo et al. [14] found similar levels of glucose in the blood serum of all the groups with and without the enzyme. However, higher doses of the enzyme increased total protein and albumin contents but reduced BUN and diamine oxidase contents. The levels of total protein and albumin in the groups supplemented with protease were similar to the values observed in the current study. Tactacan et al. [50] conducted an experiment and found that protease supplementation had no effect on the BUN level. Ma et al. [49] observed changes in blood serum parameters when different protease types were used in the diets.

### 4.3. Extrusion and Protease Effect

The combination of extrusion and protease supplementation generally did not affect the parameters under analysis, except for the blood serum GGT and the acetoacetic acid content in the cecal digesta. Further research is necessary to explain the causes of these changes.

## 5. Conclusions

To sum up, although extrusion improved the nutritional value of faba bean seeds, especially by reducing their ANF content, it did not improve the pigs’ growth performance, digestibility of nutrients, intestinal structure, and physiology when there 10% of seeds were present in the diet. The addition of protease at a dose of 0.5 g/kg to the diet with the nutrient matrix containing soybean meal and raw or extruded faba bean seeds at an amount below 10% produced similar growth results to those observed in the group in which the enzyme was not used. Further extrusion of different feed ingredients is necessary to maximise the beneficial effect of thermal treatments. It is also necessary to conduct research to determine interactions between the use of enzymes and extrusion. The supplementation of feed for pigs with protease on commercial farms using legume seeds or protein sources of poorer quality in mixtures may be economically feasible, but this problem requires further research.

## Figures and Tables

**Figure 1 animals-12-00563-f001:**
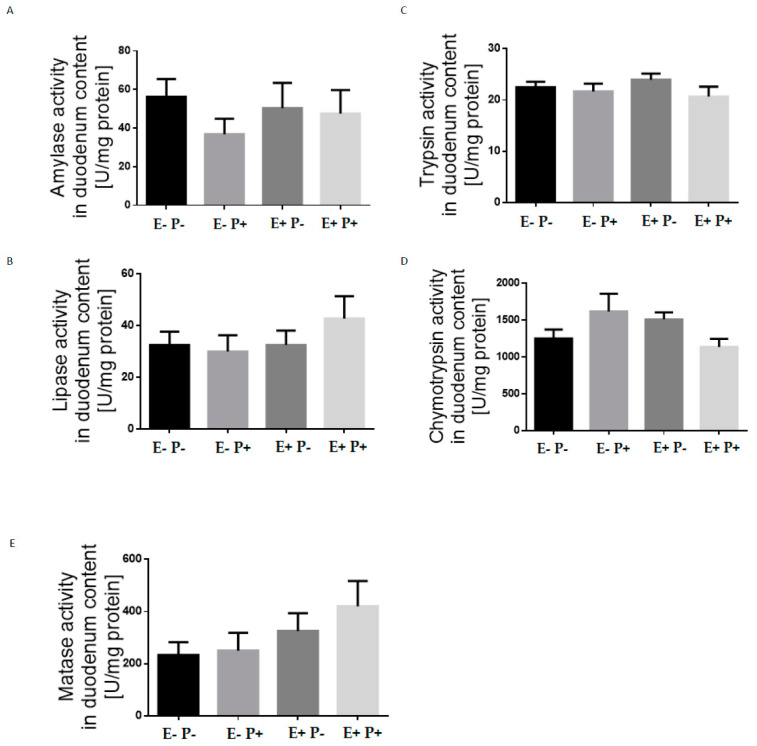
Enzyme activity (U/mg protein) in the pigs’ duodenum (*n* = 8). The effect of extrusion treatment and protease supplementation on amylase (**A**), lipase (**B**), trypsin (**C**), chymotrypsin (**D**), and maltase (**E**) activity in the duodenum content. The results are expressed as mean ± S.E.M.; ‘‘E−’’—no treatment; “P−“—“—no enzyme supplementation; ‘‘E+’’—treatment; “P+”–enzyme supplementation.

**Table 1 animals-12-00563-t001:** The chemical composition, anti-nutritional factors, and amino acid profile of raw and extruded faba bean seeds (*n* = 4).

Item (% DM)	Faba Bean		*p*
	Extrusion −	Extrusion +	
Crude protein	34.7 ± 1.6	32.7 ± 1.9	NS
ADF	11.8 ± 1.0	11.4 ± 0.9	NS
NDF	19.8 ± 1.4	12.9 ± 0.9	≤0.001
Starch	41.7 ± 4.2	41.1 ± 3.3	NS
RS	18.3 ± 0.6	0.9 ± 0.1	≤0.001
Antinutritional factors			
Oligosaccharides	3.03 ± 0.2	3.01 ± 0.1	NS
Tannins	0.006 ± 0.001	0.006 ± 0.001	NS
TIA	0.060 ± 0.001	0.020 ± 0.001	≤0.001
Phytic-P	0.50 ± 0.03	0.19 ± 0.01	0.001
Total NSP	15.36 ± 0.93	15.21 ± 0.78	NS
Essential amino acids (g/16g N)			
Lysine	5.3 ± 0.3	5.3 ± 0.4	NS
Threonine	3.1 ± 0.1	3.0 ± 0.2	NS
Methionine	0.5 ± 0.0	0.5 ± 0.1	NS
Cysteine	1.0 ± 0.0	1.0 ± 0.1	NS
Isoleucine	3.5 ± 0.4	3.5 ± 0.2	NS
Valine	4.0 ± 0.2	4.0 ± 0.1	NS
Leucine	6.5 ± 0.3	5.9 ± 0.4	NS
Phenylalanine	3.7 ± 0.2	3.7 ± 0.1	NS
Histidine	2.4 ± 0.1	2.4 ± 0.1	NS
Arginine	8.3 ± 0.5	8.2 ± 0.2	NS
Glycine	3.7 ± 0.3	3.6 ± 0.3	NS
Non-essential amino acids (g/16g N)			
Tyrosine	2.7 ± 0.1	2.6 ± 0.2	NS
Alanine	3.5 ± 0.2	3.5 ± 0.1	NS
Aspartic acid	9.4 ± 0.5	9.4 ± 0.5	NS
Glutamic acid	16.9 ± 0.9	16.8 ± 1.0	NS
Serine	4.2 ± 0.3	4.1 ± 0.4	NS
Proline	3.2 ± 0.1	3.2 ± 0.1	NS

ADF—acid detergent fibre; NDF—neutral detergent fibre; RS—resistant starch; TIA—trypsin inhibitors; NSP—non-starch polysaccharides; *p*—significance at *p* ≤ 0.05. The results are expressed as mean ± standard deviation. NS—insignificant.

**Table 2 animals-12-00563-t002:** The composition and content of nutrients in the experimental diets (%).

Treatments				
Extrusion	−	−	+	+
Enzyme	−	+	−	+
Ingredients (%)				
Soybean meal	16.300	12.600	16.300	12.600
Faba bean	9.100	9.100	9.100	9.100
Wheat	40.000	40.000	40.000	40.000
Triticale	28.065	32.715	28.065	32.715
Premix 0.5% *	0.500	0.500	0.500	0.500
Phosphate 1-Ca	1.150	1.150	1.150	1.150
Limestone	0.850	0.850	0.850	0.850
NaCl	0.350	0.350	0.350	0.350
Rapeseed oil	2.500	1.500	2.500	1.500
L-Lysine HCl	0.440	0.440	0.440	0.440
DL-Methionine	0.150	0.150	0.150	0.150
DL-Tryptophan	0.025	0.025	0.025	0.025
L-Threonine	0.270	0.270	0.270	0.270
TiO_2_	0.300	0.300	0.300	0.300
Protease	−	0.050	−	0.050
Feed nutritional value (calculated) in dry matter				
Dry matter (%)	88.60	88.92	89.10	89.49
ME (MJ/kg)	13.72	13.72	13.72	13.72
CP%	17.90	16.88	17.90	16.88
Dig. CP%	14.82	14.82	14.81	14.82
Dig. Lys%	1.150	1.150	1.150	1.150
Dig. Met%	0.363	0.363	0.363	0.363
Dig. Trp%	0.259	0.259	0.259	0.259
Dig. Thr%	0.722	0.722	0.722	0.722
Ca%	0.759	0.759	0.759	0.759
P%	0.654	0.654	0.654	0.654
Dig. P%	0.433	0.433	0.433	0.433
Na%	0.152	0.152	0.152	0.152

‘‘−’’—no treatment or enzyme supplementation; ‘‘+’’—treatment or enzyme supplementation. * The mineral and vitamin premix contained the following amounts of components per 1 kg: choline chloride—40,000 mg, Fe—15,000 mg, Cu—4000 mg, Co—60 mg, Mn—6000 mg, Zn—15,000 mg, I—120 mg, Se—30 mg, antioxidants (butylated hydroxyanisole, butylated hydroxytoluene); vitamin A—1,500,000 IU, vitamin D_3—_300,000 IU; vitamin E—10,500 mg, vitamin K_3—_220 mg, vitamin B_1—_220 mg, vitamin B_2_—600 mg, vitamin B_6_—450 mg, pantothenic acid—1500 mg, nicotinic acid—3000 mg, folic acid—300 mg, vitamin B_12_—3700 µg, biotin—15,000 µg, Ca—260 g; ME—metabolisable energy; CP—crude protein; Dig.P—digestible phosphorus, Dig.—digestible.

**Table 3 animals-12-00563-t003:** Body weight, body weight gain, daily weight gain, daily feed intake, and feed utilisation of pigs (*n* = 8).

Extrusion	−	−	+	+	*p*			
Protease	−	+	−	+	*p*	Extrusion	Protease	Interaction
IBW 0d (kg)	9.36	9.43	9.29	9.64	0.960	0.885	0.666	0.773
BW 14d (kg)	14.14	13.86	14.00	14.50	0.908	0.703	0.870	0.551
FBW 28d (kg)	23.43	22.79	24.57	24.71	0.595	0.196	0.830	0.737
DWG (kg/day)								
0–14 days	0.34	0.32	0.34	0.35	0.804	0.593	0.741	0.453
15–28 days	0.66	0.64	0.75	0.73	0.325	0.078	0.613	0.998
0–28 days	0.50	0.48	0.55	0.54	0.325	0.084	0.580	0.766
DFI (kg)								
0–14 days	0.63	0.65	0.68	0.69	0.349	0.109	0.427	0.897
15–28 days	1.05	1.12	1.22	1.27	0.203	0.048	0.420	0.892
0–28 days	0.84	0.89	0.95	0.98	0.220	0.055	0.429	0.894
FCR (kg/kg)								
0–14 days	1.85	2.09	2.08	2.06	0.454	0.419	0.368	0.285
15–28 days	1.58	1.78	1.63	1.76	0.223	0.863	0.045	0.651
0–28 days	1.67 ^b^	1.87 ^a^	1.75 ^a,b^	1.83 ^a^	0.035	0.735	0.008	0.221

The results in the table are expressed as mean values; ‘‘−’’—no treatment or enzyme supplementation; ‘‘+’’—treatment or enzyme supplementation; IBW—initial body weight; BW—body weight on the 14th day; FBW—final body weight; DWG–daily weight gain; DFI—daily feed intake; FCR—feed conversion ratio; ^a,b^—the means in the rows marked with different letters are significantly different at *p* ≤ 0.05.

**Table 4 animals-12-00563-t004:** The apparent ileal digestibility coefficients of crude protein and dry matter in the diets (*n* = 8).

Extrusion	−	−	+	+	*p*			
Protease	−	+	−	+	*p*	Extrusion	Protease	Interaction
Crude protein (%)	73.64	72.74	74.20	74.81	0.797	0.391	0.784	0.784
Dry matter (%)	76.43	77.15	77.36	77.45	0.942	0.636	0.623	0.623

The results in the table are expressed as mean values; AID—apparent ileal digestibility; ‘‘−’’— no treatment or enzyme supplementation; ‘‘+’’—treatment or enzyme supplementation; *p*—significance at *p* ≤ 0.05.

**Table 5 animals-12-00563-t005:** The physical parameters of the liver, digesta pH, ammonia concentration, dry matter content, and viscosity in the ileum and cecum (*n* = 8).

Extrusion	−	−	+	+	*p*			
Protease	−	+	−	+	*p*	Extrusion	Protease	Interaction
Liver weight (g)	735.33	748.00	757.67	831.00	0.439	0.248	0.343	0.501
Liver/BWG ratio (kg/kg)	0.032	0.031	0.032	0.030	0.828	0.756	0.417	0.756
pH								
Ileum	5.90	5.56	5.79	5.56	0.647	0.796	0.225	0.958
Cecum	5.16 ^b^	5.13 ^b^	5.42 ^a^	5.53 ^a^	≤0.001	≤0.001	0.509	0.430
Ammonia (µmol/g digesta)								
Ileum	13.81	15.71	11.96	11.18	0.443	0.144	0.791	0.529
Cecum	15.80	16.42	18.92	18.13	0.059	0.209	0.897	0.622
Dry matter (%)								
Ileum	12.28 ^a^	11.53 ^a,b^	10.66 ^b,c^	9.97 ^c^	0.02	0.004	0.161	0.958
Cecum	9.14	10.73	10.08	10.30	0.605	0.763	0.298	0.430
Viscosity (cP)								
Ileum	1.11	1.10	1.09	1.19	0.758	0.702	0.553	0.423

The results in the table are expressed as mean values; ‘‘−’’—no treatment or enzyme supplementation; ‘‘+’’—treatment or enzyme supplementation; BWG—body weight gain; ^a,b,c^—the means in the rows marked with different letters are significantly different at *p* ≤ 0.05.

**Table 6 animals-12-00563-t006:** The concentration of short-chain fatty acids (µmol/g digesta) in the pigs’ cecal digesta (*n* = 8).

Extrusion	−	−	+	+	*p*			
Protease	−	+	−	+	*p*	Extrusion	Protease	Interaction
Acetic acid	55.17 ^b^	57.88 ^a,b^	58.65 ^a^	59.68 ^a^	0.033	0.016	0.085	0.475
Propionic acid	27.17 ^a^	23.55 ^b^	23.15 ^b^	21.00 ^b^	≤0001	≤0.0001	0.003	0.376
Isobutyric acid	2.75	2.70	1.90	1.80	0.396	0.094	0.882	0.960
Butyric acid	10.50 ^a^	8.70 ^b^	6.18 ^c^	4.73 ^d^	≤0.0001	≤0.0001	≤0.0001	0.643
Isovaleric acid	1.75 ^a^	0.70 ^b^	0.62 ^b^	0.35 ^b^	≤0.0001	≤0.0001	≤0.0001	0.072
Valeric acid	2.82 ^a^	1.55 ^b^	1.08 ^b^	3.37 ^a^	≤0.0001	0.901	0.140	≤0.0001
Acetoacetic acid	0.00 ^c^	5.20 ^b^	8.58 ^a^	9.35 ^a^	≤0.0001	≤0.0001	0.011	0.060

The results in the table are expressed as mean values; ‘‘−’’—no treatment or enzyme supplementation; ‘‘+’’—treatment or enzyme supplementation; ^a,b,c,d^—the means in the rows marked with different letters are significantly different at *p* ≤ 0.05.

**Table 7 animals-12-00563-t007:** The morphometric parameters of the pigs’ ileum (*n* = 8).

Extrusion	−	−	+	+	*p*			
Protease	−	+	−	+	*p*	Extrusion	Protease	Interaction
Villus length (µm)	413.31	449.51	393.18	409.08	0.707	0.393	0.462	0.773
Villus width (µm)	104.99	108.73	104.24	105.04	0.911	0.649	0.643	0.764
Crypt depth (µm)	160.47	163.43	168.16	166.07	0.901	0.506	0.956	0.744
Villus/crypt ratio	2.56	2.77	2.35	2.49	0.562	0.250	0.420	0.875

The results in the table are expressed as mean values; ‘‘−’’—no treatment or enzyme supplementation; ‘‘+’’—treatment or enzyme supplementation; *p*—significance at *p* ≤ 0.05.

**Table 8 animals-12-00563-t008:** The pigs’ blood biochemical indexes (*n* = 8).

Extrusion	−	−	+	+	*p*			
Protease	−	+	−	+	*p*	Extrusion	Protease	Interaction
Glucose (mg/dL)	61.10 ^b^	59.19 ^b^	69.86 ^a^	63.80 ^a,b^	0.030	0.013	0.122	0.424
TG (mg/dL)	76.94	80.71	85.39	73.52	0.270	0.886	0.358	0.082
Total cholesterol (mg/dL)	91.13	84.90	92.33	87.95	0.658	0.644	0.253	0.841
Albumin (g/dL)	2.96	2.89	3.11	2.92	0.329	0.315	0.156	0.524
Total protein (g/dL)	4.75	4.69	4.69	4.54	0.335	0.411	0.400	0.687
ALT (IU/L)	5.17 ^c^	7.21 ^b^	8.72 ^a,b^	9.31 ^a^	≤0.001	0.001	0.058	0.284
AST (IU/L)	28.88	27.84	30.5	33.59	0.515	0.205	0.719	0.474
ALP (IU/L)	88.17	91.72	91.58	102.38	0.591	0.367	0.358	0.640
GGT (IU/L)	14.37 ^a^	9.21 ^a,b^	4.79 ^c^	10.32 ^a,b^	0.043	0.069	0.935	0.024
LDH (IU/L)	283.21	286.67	283.21	306.86	0.711	0.551	0.424	0.551
Creatinine (mg/dL)	0.69	0.66	0.66	0.62	0.890	0.581	0.581	0.999
BUN (mg/dL)	16.26	15.16	17.58	15.00	0.370	0.616	0.119	0.524
Urea (mg/dL)	34.8	32.44	37.62	32.01	0.370	0.616	0.119	0.524

The results in the table are expressed as mean values; ‘‘−’’—no treatment or enzyme supplementation; ‘‘+’’—treatment or enzyme supplementation; TG—triglycerides; ALT—alanine aminotransferase; AST—aspartate aminotransferase; ALP—alkaline phosphatases; GGT—gamma-glutamyl transferase; LDH—lactate dehydrogenase; BUN—blood urea nitrogen; ^a,b,c^—the means in the rows marked with different letters are significantly different at *p* ≤ 0.05.

## Data Availability

Data is available at a reasonable request to the corresponding authors.

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
