# Peer review of "The Effects of Protease Supplementation and Faba Bean Extrusion on Growth, Gastrointestinal Tract Physiology and Selected Blood Indices of Weaned Pigs"

_animals, 2022, doi:10.3390/ani12050563_

Round 1

Reviewer 1 Report

1. The title of Table 3-8 said "n=8", but all the data presented in the table is an average value? If it is the average value, please emphasize it in the table note; if not, it can also be displayed as in Table 2.

2. In Line 223, there are two full stops in the sentence"and depth of intestinal crypts (in ten replication). . ".

3. In Line 296, Fig. 1 is not presented in this manuscript, please confirm again.

4. In Line 304-305, there is a grammatical error in the sentence "There were significant differences in the glucose, ALT, and GGT levels."

5. The protease additive only increased the acetic acid content of the chyme, how can this be explained, and will it cause adverse effects on the animal body?

6. At Line 435, the citation format of the reference in the sentence "G Tactacan, et al. [502] conducted an experiment...." is incorrect.

7. After adding protease, GGT significantly decreased, but adding protease GGT increased significantly during extrusion. What is the explanation for this? In addition, does it indicate damage to the pig's liver? Did the autopsy reveal any liver lesions, and was the liver histological staining done?

8. Reference 23 format is different from other references.

9. References 26 and 30 are missing page numbers.

Author Response

We would like to thank the Reviewers for their valuable comments and suggestions, which helped us to improve our manuscript. We have revised our paper accordingly. We hope our manuscript meets their expectations and can be published in the Animals.

Reviewer 1:

  1. The title of Table 3-8 said "n=8", but all the data presented in the table is an average value? If it is the average value, please emphasize it in the table note; if not, it can also be displayed as in Table 2.

Response: Yes, all the data in the table are mean values. We have added this information below the tables.

  1. In Line 223, there are two full stops in the sentence "and depth of intestinal crypts (in ten replication). . ".

Response: Yes, it is true, It’s our mistake. We have deleted one full stop.

  1. In Line 296, Fig. 1 is not presented in this manuscript, please confirm again.

Response: Fig. 1 is included in the attachment to this manuscript. We have included it in a separate file because its quality was not satisfactory for one of the reviewers (Supplement 1. Figure_1_A-E animals 1516371)

  1. In Line 304-305, there is a grammatical error in the sentence "There were significant differences in the glucose, ALT, and GGT levels."

Response: We have checked and corrected this sentence.

  1. The protease additive only increased the acetic acid content of the chyme, how can this be explained, and will it cause adverse effects on the animal body?

Response: The protease additive increased the acetic acid content in the cecal digesta insignificantly. A tendency was only observed (P=0.085) (Table 6). We do not think this had any adverse effect on the animal parameters.

  1. At Line 435, the citation format of the reference in the sentence "G Tactacan, et al. [502] conducted an experiment...." is incorrect.

Response: Yes, it is true. It’s our mistake. We have corrected it.

  1. After adding protease, GGT significantly decreased, but adding protease GGT increased significantly during extrusion. What is the explanation for this? In addition, does it indicate damage to the pig's liver? Did the autopsy reveal any liver lesions, and was the liver histological staining done?

Response: During the autopsy we did not observe liver damage. The liver was not stained histologically. The interaction between the factors in the GGT is difficult to explain. We only suppose the highest GGT level may have been caused by a higher feed intake (but it was not confirmed statistically), which generated a greater amount of nutrients to be metabolised in the liver. But we did not include this explanation in our manuscript.

  1. Reference 23 format is different from other references.

Response: Yes, it is true, It’s our mistake. We have corrected it.

  1. References 26 and 30 are missing page numbers.

Response: We have added page numbers.

Reviewer 2 Report

Comments, review of manuscript id: Animals-1594752.

Title: ”The Effects of Protease Supplementation and Faba Bean Extrusion on the Growth, Gastrointestinal Tract Physiology and Selected Blood Indices of  Weaned Pigs”

The manuscript describes the effects of protease supplementation and extrusion of faba beans in diets wit an inclusion of 9.1% faba beans for piglets after weaning.  

The English language is in general good, but some of the sentences in the manuscript must be improved to give the reader a better understanding of what you mean.

Comments.

Simple Summary, lines 23 – 24) and Abstract (line 41 – 42). As you describe this here, it is somewhat confusing. As I read the manuscript, the reduced inclusion of soybean meal and rapeseed oil was a part of the study design and not an observed effect of the study. See your description in Material and Methods (lines 111 - 114) and in the Discussion (lines 425 – 428). With this in mind I suggest that you rewrite these line in the Simple Summary and in the Abstract.

Material and Methods, (2.. Animals, diets, protease), line 133. What do you mean by writing “…were offered their meal in a staggered fashion…”. Please explain this better.

 Material and Methods, (2.6 Chemical analysis), Later in the manuscript you refer to the content of TIA, but TIA is not described in Material and Methods.

Results, (3.1 The chemical composition of faba bean seeds), line 238. The NDF concentration in the extruded faba beans was significantly lower, how do you explain this result. You have not discussed the NDF concentration in the discussion part of the manuscript.

Results, (3.1 The chemical composition of faba bean seeds), line 241. Here you write that TIA content was significantly reduced by extrusion, but the TIA contents are not shown in Table 2. The TIA is not defined in your manuscript.

Results, (3.2 Animal experiment), Table 3. In Table 3 you show results for both “BWG, kg/period” and “DWG, kg/day”. Both of this describe the same result and it is not necessary to include both. You should only show the results for “DWG, kg/day”.  

Results, (3.2 Animal experiment), Figure 1. Figure 1 is lacking in the manuscript.   

Discussion, (4.1 Extrusion effect), line 332. The TIA is not defined or shown in Table 2, as commented earlier.

Conclusion, lines 446 – 447. Here you write “…when there was a low level of seeds in the diet.” This description is not clear and must be written better.

Author Response

Reviewer 2:

Title: ”The Effects of Protease Supplementation and Faba Bean Extrusion on the Growth, Gastrointestinal Tract Physiology and Selected Blood Indices of  Weaned Pigs”

The manuscript describes the effects of protease supplementation and extrusion of faba beans in diets with an inclusion of 9.1% faba beans for piglets after weaning. 

The English language is in general good, but some of the sentences in the manuscript must be improved to give the reader a better understanding of what you mean.

Response: The manuscript has been proofread by a native speaker.

  1. Simple Summary, (lines 23 – 24) and Abstract (line 41 – 42). As you describe this here, it is somewhat confusing. As I read the manuscript, the reduced inclusion of soybean meal and rapeseed oil was a part of the study design and not an observed effect of the study. See your description in Material and Methods (lines 111 - 114) and in the Discussion (lines 425 – 428). With this in mind I suggest that you rewrite these line in the Simple Summary and in the Abstract.

Response: We have changed the abstract and the simple summary.

  1. Material and Methods, (2.. Animals, diets, protease), line 133. What do you mean by writing “…were offered their meal in a staggered fashion…”. Please explain this better.

 Response: The information in the manuscript has been specified.

  1. Material and Methods, (2.6 Chemical analysis), Later in the manuscript you refer to the content of TIA, but TIA is not described in Material and Methods.

 Response: In the manuscript we refer to the content of TIA. TIA is an abbreviation for trypsin inhibitor. We have used TIA instead of trypsin inhibitor in all parts of the manuscript. 

  1. Results, (3.1 The chemical composition of faba bean seeds), line 238. The NDF concentration in the extruded faba beans was significantly lower, how do you explain this result. You have not discussed the NDF concentration in the discussion part of the manuscript.

 Response: Yes, it is true. We did not discuss the NDF concentration in the Discussion section. We have explained this result.

  1. Results, (3.1 The chemical composition of faba bean seeds), line 241. Here you write that TIA content was significantly reduced by extrusion, but the TIA contents are not shown in Table 2. The TIA is not defined in your manuscript.

 Response: TIA is an abbreviation for trypsin inhibitor. We have used TIA instead of trypsin inhibitor in all parts of the manuscript. 

  1. Results, (3.2 Animal experiment), Table 3. In Table 3 you show results for both “BWG, kg/period” and “DWG, kg/day”. Both of this describe the same result and it is not necessary to include both. You should only show the results for “DWG, kg/day”.  

 Response: As suggested, we have deleted “BWG, kg/period” both from Table 3 and the whole manuscript.

  1. Results, (3.2 Animal experiment), Figure 1. Figure 1 is lacking in the manuscript.   

 Response: Fig. 1 is included in the attachment to this manuscript. We have included it in a separate file because its quality was not satisfactory for one of the reviewers - Supplement 1. Figure_1_A-E animals 1516371

  1. Discussion, (4.1 Extrusion effect), line 332. The TIA is not defined or shown in Table 2, as commented earlier.

Response:  We have changed it.

  1. Conclusion, lines 446 – 447. Here you write “…when there was a low level of seeds in the diet.” This description is not clear and must be written better.

Response: We have changed this sentence.

Round 2

Reviewer 2 Report

The corrections done by the authors of the manuscript "The Effects of Protease Supplementation and Faba Bean Extrusion on the Growth, Gastrointestinal Tract Physiology and Selected Blood Indices of Weaned Pigs" are done according to the comments from the reviewers. The only comment to the revised manuscript is a writing error in "Figure 1. "Matase activity" is spelled wrong, the correct is "Maltase acrivity".

This manuscript is a resubmission of an earlier submission. The following is a list of the peer review reports and author responses from that submission.

Round 1

Reviewer 1 Report

The paper titled "The Effects of Protease Supplementation and Faba Bean Extrusion on the Growth, Gastrointestinal Tract Physiology and Selected Blood Indices of Weaned Pigs" aims to explore how to affect animal growth parameters, digestibility of nutrients, selected physiological parameters of the digestive tract, and biochemical blood parameters with the extrusion of broad bean seeds and/or the addition of proteases to the diet of pigs. It was found that extrusion and protease increased the content of acetic acid and acetoacetic acid in the cecal chyme, but the extrusion and protease additives did not improve the growth performance of pigs. The method and thinking of this article are clear, the writing is fluent, but the result is less narrated. The following questions and suggestions are put forward for this research.

Major problem:

  1. Please provide the H&E staining figure and scanning electron microscope of the ileum.
  2. Line 72-75, some studies have found that "found that the protease additive could improve the performance of fattening pigs fed with faba bean seeds", but why does this article not improve the growth performance of pigs with protease additives?
  3. Since extrusion and protease additives did not improve the growth performance of pigs, please clarify what instructions and significance this research has for future work.
  4. Please explain how extrusion reduces the level of ANFs in broad beans?

Minor:

  1. Please provide the age of the experimental animal.
  2. Please explain why we chose piglets instead of adult pigs as the experimental animals in this experiment
  3. Please add the n value of each experimental animal in 2.1. Faba bean seeds to facilitate readers' understanding.
  4. Have all the results analysis and processing been tested for normal distribution?
  5. Please add the significant difference marks in the A-F histogram in Fig.1.
  6. Is there any necessary relationship between the digestive enzyme activities in the chyme of the duodenum and the shape of the ileum?
  7. Why is there no correlation analysis of animal growth parameters, nutrient digestibility, biochemical blood parameters, cecal parameters and enzyme activity?
  8. Please add a small summary of some results to the results to facilitate readers' understanding.
  9. Some references are incomplete, such as 11, 23, 24, 26, and 30.
  10. More than half of the documents have been published too long ago. Please replace some of the documents with documents from the past five years.

Author Response

Overall comment:

We would like to thank you for the review.

A detailed response to the Reviewers

Major problem:

  1. Please provide the H&E staining figure and scanning electron microscope of the ileum.

The morphometric analysis was commissioned for a fee from another University. The contractor's consultation proved a mistake in the methodology—specimens stained with the PAS (Periodic acid-Schiff) technique using the Schiff reagent for intestinal morphometric analysis. A Nikon Ci-L microscope integrated with a NikonDS-Fi3 camera and NIS Elements software (Nikon Instruments Inc.) was used to measure the height and width of villi, depth of intestinal crypts, and thickness of the muscular layer. The correct was added in section 2.7.

  1. Line 72-75, some studies have found that "found that the protease additive could improve the performance of fattening pigs fed with faba bean seeds", but why does this article not improve the growth performance of pigs with protease additives?

Yes, it is true. In the article by Swiatkiewicz et al. (2018), protease did not improve the growth performance of pigs. Other literature should be cited in this part of the introduction. We have added in the text (Hanczakowska, E.; Swiatkiewicz, M. Legume seeds and rapeseed press cake as replacers of soybean meal in feed for fattening pigs. Ann. Anim. Sci. 2014, 14, 4, 921).

  1. Since extrusion and protease additives did not improve the growth performance of pigs, please clarify what instructions and significance this research has for future work.

We added a simple summary and some information in lines 25-27.

  1. Please explain how extrusion reduces the level of ANFs in broad beans?

We added a part to explain how extrusion reduces the level of ANFs:  L. 337-342

Minor:

Please provide the age of the experimental animal.

It was added in the text.

Please explain why we chose piglets instead of adult pigs as the experimental animals in this experiment

We chose piglets instead of adult pigs in this experiment, because in actual literature, there is little information about using new varieties of faba bean seeds in this group of animals.

Please add the n value of each experimental animal in 2.1. Faba bean seeds to facilitate readers' understanding. –

We changed and added in sections 2.1 and 2.2 .

Have all the results analysis and processing been tested for normal distribution?

Yes, all the results analysis and processing have been tested for normal distribution.

Please add the significant difference marks in the A-F histogram in Fig.1.

Neither the extrusion nor the protease supplementation affected (P > 0.05) the activity of enzymes in the distal colon; that is why we did not add significant difference marks in Figure 1.

Is there any necessary relationship between digestive enzyme activities in the duodenum chyme and shape of the ileum?

There is no relationship between digestive enzyme activities in the chyme of the duodenum and the shape of the ileum. No significant differences were found between the factors, and we have described them together.

Why is there no correlation analysis of animal growth parameters, nutrient digestibility, biochemical blood parameters, cecal parameters, and enzyme activity?

In our study, we established two factors, which we focused on in the statistical analysis and the interactions between the studied factors. We did not check the correlation between the variables. We do not consider it essential in this work, and other reviewers have not indicated it to. If it is necessary, we can make this analysis.

Please add a small summary of some results to the results to facilitate readers' understanding.

We added a simple summary-Line 16-27.

Some references are incomplete, such as 11, 23, 24, 26, and 30.

We checked and completed in references.

More than half of the documents have been published too long ago. Please replace some of the documents with documents from the past five years.

Yes, it is true, some documents have been published long ago, but the old literature is mostly about methodology. We have revised some documents and replaced them for the past five years.

Our comments/information’s:

We hope our answers have cleared up all the confusion. We would be pleased to publish the revised manuscript in MDPI Animals.

Yours sincerely

Małgorzata Kasprowicz-Potocka

Reviewer 2 Report

Major comments.

There are several problems with respect to Table 1.

First of all, if you want to compare the effect of adding protease, you should make a comparison with the same basal diet and added protease. In the case of both diets, there is one point less soybean cake and more triticale content. Is the effect you see between the two diets due to protease or those changes?

There is also a concept that I do not quite understand. The crude protein content is approximately one point between the experimental diets, however both diets (with different protein raw materials) are added the same amount of synthetic amino acids, and in the end they have the same amount of digestible amino acids. How can it be that with different formulas and adding the same amount of synthetic amino acids we obtain such a similar chemical composition? According to the approach, I understand that only comparisons could be made between the treatment "raw faba bean and extruded faba bean" as well as "raw faba bean + protease and extruded faba bean + protease".

The data is calculated (which already gives us doubts) but also the units are a bit strange, are they calculated fresh or dry? There is no data on the humidity of the feed, without this data we cannot correctly specify the contribution of the feed.

Table 2. I understand that they are the nutritional composition of the two types of faba, however, there is a statistical analysis. What is this about?

Have you made daily growths with 8 animals only and conversion rates?

Minor comments.

Review the format of the tables. Definition as “The pig’s growth performance (n = 8) is completely imprecise.

Some graphics are pixelated

Author Response

Reviewer 2

Thank you very much for Your review.

A detailed response to the Reviewers

Major comments.

  1. There are several problems with respect to Table 1.

First of all, if you want to compare the effect of adding protease, you should make a comparison with the same basal diet and added protease. In the case of both diets, there is one point less soybean cake and more triticale content. Is the effect you see between the two diets due to protease or those changes?

Our response: Yes, in groups no. 2 and 4, part of the soybean meal was replaced by triticale. This such procedure is normally performed in mixtures in which diet is based on the nutrient matrix for enzymes as in our research. According to the producer's own data, the diet was specially prepared for this study by protease producer based on the basal diet composition and enzyme activity.

Yes, the effect we see between the two diets is due to the addition of an enzyme protease.

No, the possible differences could not be explained by the reduction of soybean because the enzyme “added” nutrient content in the mixture. In matrix, we calculate the future effects of enzyme activity.

  1. There is also a concept that I do not quite understand. The crude protein content is approximately one point between the experimental diets, however both diets (with different protein raw materials) are added the same amount of synthetic amino acids, and in the end they have the same amount of digestible amino acids. How can it be that with different formulas and adding the same amount of synthetic amino acids we obtain such a similar chemical composition? According to the approach, I understand that only comparisons could be made between the treatment "raw faba bean and extruded faba bean" as well as "raw faba bean + protease and extruded faba bean + protease".

Yes, it is possible, that’s why calculated values are provided. Examples of enzymes and matrices are very well described in the publication: Bedford, M. R., Cowieson, A. J. (2020). Matrix values for exogenous enzymes and their application in the real world. Journal of Applied Poultry Research, 29(1), 15-22. When we add an enzyme, we expect better digestibility of nutrients to give to the diet a little amount of some nutrients as protein when we use protease. We do not change the crystal amino acids levels because we concentrated only high protein components. As we pointed out before, the producer prepared the protease-diets based on their own data contacting with this enzyme effectivity. It was based on real solutions used in the practice.

  1. The data is calculated (which already gives us doubts) but also the units are a bit strange, are they calculated fresh or dry? There is no data on the humidity of the feed, without this data we cannot correctly specify the contribution of the feed.

The data is calculated in dry matter. It was added dry matter content in all the experimental feed mixtures.

  1. Table 2. I understand that they are the nutritional composition of the two types of faba; however, there is a statistical analysis. What is this about?

Based on the literature, we know that the temperature, pressure, and conditioning during extrusion cooking can change the chemical composition of feeds. In manuscript (Table 2) ​we present the nutritional composition of the two types of faba bean extruded and not extruded because we use both in the feed mixtures  (n=4), and student’s t-test was applied to compare the chemical composition of raw and extruded faba bean seeds (information is provided in the M&M section).

  1. Have you made daily growths with 8 animals only and conversion rates?

In one dietary treatment (group) we made 8 observations (replications). Animals were kept in individual cages.

Minor comments.

Review the format of the tables. Definition as “The pig’s growth performance (n = 8) is completely imprecise.

We agree and we changed a definition in table 3.

Some graphics are pixelated

We added new graphics in the supplement material.

Our comments/information’s:

We hope our answers have cleared up all the confusion. We would be pleased to publish the revised manuscript in MDPI Animals.

Yours sincerely

Małgorzata Kasprowicz-Potocka

Round 2

Reviewer 1 Report

This article is written smoothly, the materials and methods are detailed and complete, the result data is complete, and the discussion echoes the results. And most of the problems are complete, explained and corrected. So I suggest to receive this article.

Reviewer 2 Report

Thanks for the modifications but I keep thinking this: 

As various raw materials are modified, the digestibility not only of the protein but of other nutritional components is modified. The observed differences cannot be assumed only to the effect of the enzymatic activity. 

8 animals per treatment is not enough individuals for a growth experiment.